



# Stepping beyond perfectly mixed conditions in soil hydrological modelling using a Lagrangian approach

Alexander Sternagel[1], Ralf Loritz[1], Brian Berkowitz[2], Erwin Zehe[1]

[1] Karlsruhe Institute of Technology (KIT), Institute of Water Resources and River Basin Management, Hydrology, Germany
[2] Department of Earth and Planetary Sciences, Weizmann Institute of Science, Israel

*Correspondence to*: Alexander Sternagel (alexander.sternagel@kit.edu)

**Abstract.** A recent experiment of Bowers et al. (2020) revealed that diffusive mixing of water isotopes ($\delta^2$H, $\delta^{18}$O)
over a fully saturated soil sample of a few centimetres in length required several days to equilibrate completely. In
this study, we present an approach to simulate such time-delayed diffusive mixing processes on the pore scale
beyond instantaneously and perfectly mixed conditions. The diffusive pore mixing (DIPMI) approach is based on
a Lagrangian perspective on water particles moving by diffusion over the pore space of a soil volume and carrying
concentrations of solutes or isotopes. The idea of DIPMI is to account for the self-diffusion of water particles
across a characteristic length scale of the pore space using pore-size-dependent diffusion coefficients. The model
parameters can be derived from the soil-specific water retention curve and no further calibration is needed. We
test our DIPMI approach by simulating diffusive mixing of water isotopes over the pore space of a saturated soil
volume using the experimental data of Bowers et al. (2020). Simulation results show the feasibility of the DIPMI
approach to reproduce measured mixing times and concentrations of isotopes at different tensions over the pore
space. This result corroborates the finding that diffusive mixing in soils depends on the pore size distribution and
the specific soil water retention properties. Additionally, we perform a virtual experiment with the DIPMI approach
by simulating mixing and leaching processes of a solute in a vertical, saturated soil column and comparing results
against simulations with the common perfect-mixing assumption. Results of this virtual experiment reveal that the
frequently observed steep rise and long tailing of breakthrough curves, which are typically associated with non-
uniform transport in heterogeneous soils, may also occur in homogeneous media as a result of imperfect subscale
mixing in a macroscopically homogeneous soil matrix.

## 1 Introduction

Water isotopes are used widely as tracers to investigate a variety of hydrological processes (Sprenger et al., 2016).
While they were originally used to separate pre-event and event water contributions to storm runoff (Bonell et al.,
1990; Sklash et al., 1996), they are now more frequently considered as a continuous source of information to infer
travel time distributions of water through hydrological systems (e.g., McGlynn et al., 2002; McGlynn and Seibert,
2003; Weiler et al., 2003; Klaus and McDonnell, 2013). Early analyses often relied on time invariant transfer
functions, whereas some of the more recent approaches are time-dependent and, for example, use age ranked
storage as a "state" variable in combination with StorAge Selection (SAS) functions for stream flow and
evapotranspiration to infer their respective travel time distributions (Harman, 2015; Rodriguez and Klaus, 2019;
Rodriguez et al., 2021). This inference of transit times from water isotopes commonly implies a distinct relation
between water age and its isotopic composition.



However, recent laboratory and field experiments suggest that this relation and the fate of water isotopes in the soil-plant-atmosphere system may be in fact more complex than frequently assumed. Mennekes et al. (2021), for example, used in-situ probes to measure isotopic signatures of water in soil and tree xylem, during tracer irrigation experiments on the plot scale, and discussed that travel times of water fractions in soils and plants may be distinctly

different. This is in line with findings of Benettin et al. (2021), who performed lysimeter experiments with isotopic labelled waters to not only close, but also trace, all fluxes in the water balance. They found that the isotopic composition of transpiration fluxes was significantly different compared to breakthrough fluxes in soil drainage. On the pore scale, Orlowski and Breuer (2020) investigated how the isotopic composition of water depends on the retention characteristic of a soil. Their experimental results highlight "*a need to better characterize processes that*

*govern isotope fractionation with respect to soil water retention characteristics*" because they found fractionation of $\delta^2$H and $\delta^{18}$O isotopes during their diffusive movement over different pore sizes, especially under high tensions in small pores. In this context, Bowers et al. (2020) performed an experiment using a combination of extraction methods to sample isotopically defined water fractions from a saturated soil sample over the complete water retention curve to explore how fast water isotopes ($\delta^2$H, $\delta^{18}$O) mix diffusively over the entire pore size distribution.

They showed that mixing and fractionation processes of water isotopes depend on different tensions at which water is held in pores of different size. The most interesting insight of the experiment was that the isotope tracer required up to 3-4 d until it was distributed uniformly over the entire pore space, even though the studied soil sample was only a few centimetres in length.

In particular, the experimental findings of Bowers et al. (2020) suggest that ignoring self-diffusion processes of

water isotopes within the soil pore space can result in incorrect estimates of water ages and travel times, which emphasizes the requirement to include these pore scale processes into soil hydrological models. Common soil hydrological models average over pore-size-dependent differences in the flow field and concentration gradients in control volumes (Berkowitz et al., 2016) to describe diffusive mixing of water and solutes. This implies that incoming "new" event water and "old" pockets of pre-event water in soil mix perfectly and instantaneously over

the subscale pore size distribution in a single time step. This common perfect-mixing assumption is thus not in line with the recent experimental findings (e.g., Bowers et al., 2020; Orlowski and Breuer, 2020). Further studies also have shown that these different pockets of water may indeed co-exist, even in close spatial distances, without perfect mixing as well as that water and solutes repeatedly travel through the same pathways (memory effect), even after several infiltration cycles of new precipitation (Gouet-Kaplan and Berkowitz, 2011; Kapetas et al.,

2014). In this way, the establishment of stable water pockets in soils is possible, which may comprise significantly different isotopic and chemical compositions depending on the properties of infiltrating water. This imperfect mixing of water and solutes in the pore space is frequently discussed rather in the context of rapid preferential flow in macroporous structures (Beven and Germann, 1982; Beven and Germann, 2013), which is also commonly assumed to be the main reason for the characteristic steep rise and long tailing of corresponding breakthrough

curves (e.g., Berkowitz et al., 2006; Edery et al., 2014). Based on the findings of Bowers et al. (2020), we hypothesize that imperfect mixing of water and solutes over the pore sizes of a macroscopically homogeneous and saturated soil matrix will also yield such typical shapes of breakthrough curves, even without the presence of macroporous soil structures.

To account for subscale diffusive mixing of solutes or water isotopes over pore sizes in line with the findings of

Bowers et al. (2020), we propose that recent particle-based Lagrangian approaches (e.g., Berkowitz et al., 2006;



Zehe and Jackisch, 2016; Jackisch and Zehe, 2018; Engdahl et al., 2017; Engdahl et al., 2019; Schmidt et al., 2019) offer a series of new possibilities in this regard. Zehe and Jackisch (2016) showed that the conceptualization of fluid flow in partially saturated soils as a Lagrangian, advective-diffusive random walk of water particles is feasible to reproduce successfully observed soil water dynamics and distinguish explicitly pre-event and event waters. The

key was to account for a variable, pore-size-dependent mobility of water particles, which was achieved by discretizing the pore space into pores of different sizes with specific hydraulic conductivities and water diffusivities (cf. Sect. 2.1). In follow-up studies (Sternagel et al., 2019; Sternagel et al., 2021), we extended this model approach and developed the Lagrangian Soil Water and Solute Transport (LAST) Model for simulations of (reactive) solute transport combined with water motion in heterogeneous, partially saturated 1-D soil domains. These former

versions of the LAST-Model, however, assumed instantaneous, perfect mixing of solutes among water particles in a control volume, which implied that the model may smoothed out concentration gradients too fast (Sternagel et al., 2021).

In this study, we eliminate this perfect-mixing assumption and introduce the **di**ffusive **p**ore **mi**xing (DIPMI) approach to provide a Lagrangian method to improve our ability to describe diffusive mixing processes on the

pore scale. The idea of DIPMI is to account for the self-diffusion of water particles across a characteristic length scale of the pore space using pore-size-dependent diffusion coefficients. Its model parameters can be derived from the soil-specific water retention curve and no further calibration is needed. We initially test the DIPMI approach by simulating the experiment of Bowers et al. (2020) using the respective dataset. Further, we implement the DIPMI approach into our LAST-Model framework and perform a virtual experiment to test our hypothesis. To

this end, we simulate diffusive mixing and breakthrough of a representative solute in a vertical 1-D soil column during steady state, saturated flow and compare the results to simulations using the perfect-mixing assumption.

## 2 Lagrangian approach for soil-hydrological and subscale diffusion processes

### 2.1 Underlying concept of the LAST-Model

The Lagrangian perspective describes a mobile observer travelling along the trajectory of a fluid particle through

a system (Currie, 2016). As mentioned above, we have applied the Lagrangian perspective before in our LAST-Model (Sternagel et al., 2019; Sternagel et al., 2021) to describe vertical displacement of water particles with related (reactive) solute transport in interacting domains of soil matrix and macropores. Water particles are defined discretely by constant water mass and volume. They additionally carry time-dependent information about, e.g., their vertical position in both domains and solute concentrations. The two flow domains of soil matrix and

macropores are vertically subdivided into layers. This vertical discretization is required to quantify and translate the number of water particles, in combination with the water particle mass and density, into a soil water content per vertical soil layer. The soil water content in turn corresponds to the sum of volume fractions of soil water, which are stored in soil pores of different sizes. Water particles travel at different velocities in these pore fractions that are characterized by the shape of the water diffusivity and hydraulic conductivity curve. These curves are

partitioned into a certain number of pore size classes or bins ("binning") between the residual and saturated water content. Depending on the pore size class/bin in which a water particle is located, it experiences different displacements in the vertical direction by means of pore-size-specific advection and diffusivity, i.e., water particles in smaller pores experience a smaller vertical displacement step than in coarser pores. Hence, this approach



accounts for the combined effects of gravity and capillarity on water flow in partially saturated soils, as well as the subscale variability of flow velocities across different pore sizes (Zehe and Jackisch, 2016). However, in former versions of LAST (Sternagel et al., 2019; Sternagel et al., 2021), we assumed that the timescale for diffusive mixing is smaller than the simulation time step and hence, solutes perfectly and instantaneously mix over all pore

size classes/bins in a soil layer. Thus, after the non-uniform, vertical movement of particles, solute concentrations were averaged over all present water particles within a vertical soil layer per time step (perfect-mixing assumption). Furthermore, the LAST-Model allows for the simulation of sorption and degradation processes during the transport of reactive substances. Non-linear adsorption and desorption processes are realised by an explicit transfer of dissolved solute mass between water particles and surrounding sorption sites of the soil phase in a certain depth.

Adsorbed solute masses are then degraded by means of first-order kinetics.

Previous test simulations revealed that the LAST-Model effectively describes solute concentration profiles and leaching behaviours of both conservative tracers and reactive substances, on the plot and field scale under various flow conditions. In particular, the structural macropore domain of LAST represents an asset to capture the typical pattern of preferential bypassing of solutes in macroporous soils. Despite these promising results, we also showed

that our former assumption of perfect mixing of solutes within a vertical soil layer was a strong simplification that could lead to smoothing of pore-size-dependent differences in the flow field and concentration gradients (over-mixing) (Sternagel et al., 2021).

**2.2 The DIPMI approach: concept to represent subscale diffusion in a Lagrangian model**

In this study, we step beyond the use of the perfect-mixing assumption by developing a Lagrangian approach to

simulate self-diffusive mixing of water and solutes over the pore space. We explain this diffusive pore mixing (DIPMI) approach based on the schematic sketch in Figure 1:

The rectangle in the left box at $t0$ schematically illustrates a control volume with height $dz$ (e.g., a soil layer) of a 1-D vertical soil profile with total depth $z$. The width of this rectangle illustrates the entire subscale extent of pore

space $L_D$ in which fluid particles can move by self-diffusion. $L_D$ represents a characteristic flow length in the pore space, which is related to tortuosity of flow paths, the subscale distribution of pore sizes and thus, to the soil-specific water retention curve (see right box at $t0$). The extent of pore space $L_D$ and the soil water retention curve are subdivided equally into a certain number $N$ of bins $i$, which represent water storage in different pore size classes with corresponding matric potentials $\psi$. $N$ generally depends on soil-specific properties (Talbot and Ogden, 2008;

Ogden et al., 2017). We assign $N = 200$, which is suggested by numerical experiments (e.g. Zehe and Jackisch, 2016). This value of $N$ is in line with Talbot and Ogden (2008), who used a similar method and suggested that the soil moisture domain of most soil types can be discretized sufficiently by 200 bins.

Based on the Young-Laplace equation and the subdivided ("binned") soil water retention curve, we can determine the total subscale extent of the pore space $L_D$ (L) by the integral (Eq. 2) over the corresponding distribution of pore

radii $r_i$ (Eq. 1):

$$r_i = \frac{-2 \cdot \sigma}{g \cdot \rho \cdot \psi(i)},$$
Eq. 1





$$L_D = \int_{i=1}^{N} r_i \,,$$

Eq. 2

where $r_i$ (L) is the radius of a pore size class, $\sigma$ (F L$^{-1}$) the surface tension of fluid, $g$ (L T$^{-2}$) the gravitational acceleration of the earth, $\rho$ (M L$^{-3}$) the fluid density, $\psi(i)$ (L) the matric potential of a pore size class/bin, derived

from the soil water retention curve.

Each bin is defined by a constant width $\delta L_D = L_D \cdot N^{-1}$ (L) and a corresponding location within $L_D$. In our example, each bin is saturated by fluid particles carrying two different isotopic signatures (illustrated by the green and dark yellow particles). These fluid particles have a position within $L_D$ and accordingly, they are located within a certain bin ($i = 1…N$). This means that, at $t0$, the pore space is filled by fluid particles with different isotopic signatures:

coarse and medium pore size classes/bins (on the left) are filled by particles with one isotopic signature (green particles), while small pore size classes/bins (on the right) are filled by particles with another isotopic signature (dark yellow particles). Hence, fluid particles with different isotopic signatures are distinctly unmixed in the pore space before self-diffusive mixing starts at $t1$.

At $t1$, self-diffusive mixing of particles with different isotopic signatures begins. For each particle, a displacement step $\Delta d_{L_D}$ (L) along $L_D$ is calculated using a random walk equation (Eq. 3), which is then subtracted from the current position of the particle.

$$\Delta d_{L_D} = Z \cdot \sqrt{2 \cdot D(i) \cdot dt} - \left(\frac{\delta D(i)}{\delta L_D}\right),$$

Eq. 3

where $Z$ [-1,1] is a random number drawn from a standard normal distribution, $D(i)$ (L$^2$ T$^{-1}$) the diffusion coefficient in a certain bin or pore size class and $dt$ (T) the time step. The last term $\frac{\delta D(i)}{\delta L_D}$ is a correction term to avoid artefacts in case of spatially variable diffusion coefficients (see explanation below).

The random number between -1 and 1 enables the displacement of particles in the positive as well as in the negative

direction along $L_D$, representing the undirected process of molecular self-diffusion due to Brownian motion. In this diffusion process, particles are not displaced by the same diffusion coefficient $D$. Depending on the bin or pore size class in which a particle is located, it experiences a specific diffusivity. Each bin/pore size class has its own $D(i)$ value, which is determined by its proportion on the total soil porosity (proportion factor), multiplied with the molecular diffusion coefficient of free water $= 2.272 \cdot 10^{-9}$ m$^2$ s$^{-1}$ (after Mills, 1973) (Eq. 4):


$$D(i) = 2.272 \cdot 10^{-9} \cdot \left(\frac{\theta(i)-\theta_r}{\phi}\right),$$

Eq. 4

where the proportion factor comprises the respective water content of a specific bin $\theta(i)$ (-) according to the binning of the water retention curve (cf. Fig. 1), the residual water content $\theta_r$ (-) and the total soil porosity $\phi$ (-).

In this way, larger pores/bins have higher $D$ values and thus, particles experience a larger diffusive displacement in these pores/bins, while particles in smaller pores/bins with lower $D$ values experience a smaller diffusive displacement. This reflects the decline of the free path length for Brownian motions in smaller pores. With this approach of variable, pore-size-dependent diffusion coefficients, we account for the general controls on the





diffusion rate in soil solution, e.g., the diffusion coefficient of a certain fluid or substance, pore size or water content and tortuosity of flow paths (Chou et al., 2012). With these variable diffusivities in pore size classes, we add a correction term $\frac{\delta D(i)}{\delta L_D}$ (Zehe and Jackisch, 2016) to the random walk equation (Eq. 3) to avoid artificial particle accumulation in the smallest pores/bins, as stated by Uffink (1990). The random number $Z$ in Eq. 3 is either positive

5    or negative and determines in this way in which direction a fluid particle is displaced along $L_D$, i.e., if it moves diffusively in the direction of smaller or coarser pore size classes/bins (minus-sign: in direction of coarse pores, plus-sign: in direction of small pores). At the same time, the correction term has a constant negative-sign and thus, the diffusive displacement steps $\Delta d_{LD}$ of particles in the direction of coarse pores are enhanced, while they are diminished in the direction of smaller pores. In this way, calculated displacement step lengths of particles are

10    corrected with different strength. The overall, greater displacement lengths of particles in coarser pores (due to higher $D$ values) in the direction of smaller pores are balanced and, consequently, artificial particle accumulations in the smallest pores are prevented. According to its displacement step, a particle is assigned a new position within $L_D$, and if $\Delta d_{LD} > \delta L_D$, the particle is also assigned a new bin number. At the left and right boundary of the entire pore space $L_D$, particles are reflected into the pore space to avoid particle accumulation at the boundaries.

Finally, after a certain mixing time $t2$, the pore space $L_D$ in our example has reached a final equilibrium state with an uniform isotopic signature in all bins. The subscale separation ("binning") of the pore space allows for the calculation of mixing concentrations in single bins or tension areas (= certain number of adjacent bins/pore size classes within defined ranges of matric potentials).

**Figure 1.** Schematic sketch of the diffusive pore mixing (DIPMI) approach. See descriptions of Eqs. 1-4 in Sect. 2.2 for further information on the parameters.





### 3 Testing the DIPMI approach

We test our DIPMI approach by simulating the experiment of Bowers et al. (2020) with diffusive mixing of water isotopes over the pore space of a fully saturated soil volume. Furthermore, we perform a virtual experiment by simulating mixing and leaching processes of a representative solute in a vertical, saturated soil column, and

compare results of the DIPMI approach against simulations that employ the common perfect-mixing assumption.

### 3.1 Simulating the experiment of Bowers et al. (2020)

#### 3.1.1 Original experiment

Bowers et al. (2020) used a combination of extraction methods to quantify time-dependent mixing of different water isotopes ($\delta^2$H, $\delta^{18}$O) held at different tensions in fully saturated soil samples over the entire water retention

curve. Their objective was to analyse how separate soil water fractions, stored in different pore sizes, interact by self-diffusion. They took oven-dried, homogenized soil samples (18-30 g) of a sandy loam (cf. Tab. 1) and initially wetted them with isotopically light water ($\delta^2$H = -130 ‰, $\delta^{18}$O = -17.6 ‰) to a relative saturation of about 16-17 %. This water fraction represented an initial water content stored at high matric potentials in the smallest pores. The remaining free pore space was then completely saturated with isotopically heavy water ($\delta^2$H = -44 ‰, $\delta^{18}$O =

-7.8 ‰) representing new incoming water. Soil samples were then placed into horizontal cylinders and different equilibration time periods of 0 h, 8 h, 1 d, 3 d, 7 d were applied to enable mixing of the two isotopically distinct waters over the pore space by pure self-diffusion (no advection). After each time period, soil water samples were sequentially extracted from the soil samples at three subsequent tensions: (i) centrifugation at ~ < 0.016 MPa for waters in low-tension areas, (ii) centrifugation at ~ 0.016 – 1.14 MPa for waters in mid-tension areas, and finally

(iii) cryogenic vacuum distillation (CVD) at 100 MPa to capture residual waters in high-tension areas (~ > 1.14 MPa). Isotopic compositions of extracted water samples were then analysed to assess differences between the diffusive mixing behaviour in the three tension areas over 7 d. All experimental data with detailed information about soil hydraulic properties of the sandy loam are freely accessible via an Open Science Framework (Bowers and Mercer, 2020).

#### 3.1.2 Simulation with the DIPMI approach

For our simulations with the DIPMI approach, we assume a representative, saturated soil layer volume with the same soil hydraulic properties and soil water retention characteristics as the sandy loam used by Bowers et al. (2020) (see also the water retention curve in Bowers et al. (2020)). This soil layer is only defined by the extent of the pore space $L_D$ (cf. Sect. 2) and has no vertical extent, as we simulate pure diffusion over the pore space without

any vertical displacement of particles in this case. For saturation of the pore space, we generally use the same saturation procedure but we do not use the pure isotopically light and heavy waters as in the experiments (cf. Sect. 3.1.1). Instead, we use the upper and lower standard deviation (SD) values of isotopic concentrations, which Bowers et al. (2020) measured after the first extraction time at 0 h. This is necessary to enable an equal initial condition of isotopic concentrations in simulations compared to observation; we thus perform simulations with

differing initial isotopic values for light and heavy water (cf. Tab. 1). First, we use the upper SD values for light water ($\delta^2$H = -79 ‰, $\delta^{18}$O = -9.3 ‰) and heavy water ($\delta^2$H = -46 ‰, $\delta^{18}$O = -7.2 ‰). Second, we use the lower SD values for light water ($\delta^2$H = -99 ‰, $\delta^{18}$O = -12.3 ‰) and heavy water ($\delta^2$H = -48 ‰, $\delta^{18}$O = -7.8 ‰). After both




simulation runs, isotopic concentrations are averaged over the respective tension areas at each time point, resulting in the mean and SD values of our simulations (cf. Tab. 2). We use given experimental data of the soil water retention curve and Eqs. 1 and 2 to quantify the total extent of pore space $L_D$ of the soil volume by 21,000 µm and subdivide it into 200 bins or pore size classes (cf. Sect. 2.2). Additionally, we repeat simulations with (i) constant

diffusivity $D$ of $2.272 \cdot 10^{-9}$ m² s$^{-1}$ (diffusion coefficient of free water) in all bins, and (ii) linearly pore-size-distributed diffusivities $D$ over all bins calculated by Eq. 4 (cf. Sect. 2.2). Pore-size-distributed $D$ values thus range from $\sim 1.9 \cdot 10^{-9}$ m² s$^{-1}$ in bin 1 (= largest pore in low-tension area) to $\sim 9.0 \cdot 10^{-12}$ m² s$^{-1}$ in bin 200 (= smallest pore in high-tension area). We do not distinguish between specific diffusion coefficients for $\delta^2$H and $\delta^{18}$O, as Hasegawa et al. (2021) recently found generally equal diffusion properties of both isotopes in artificial and natural

porous media.

We use a total of $10^5$ particles, which corresponds to 500 particles per bin at full saturation. A high number of particles is needed to enable a stochastically valid random walk process (cf. Eq. 3) (Zehe and Jackisch, 2016). Initially, all particles are distributed randomly over all bins in $L_D$ and thus, each particle is assigned an exact position and bin number within the pore space prior to the start of the mixing process. To saturate the pore space

with the two isotopically distinct waters, we further initially define the particles in each bin that contain the isotopically light or heavy water concentrations. According to the binning of the water retention curve (cf. Sect. 2.2), we identify that the bins 168-200 correspond to a relative saturation of 16-17 % (cf. Sect. 3.1.1). Thus, these bins are filled with particles carrying the light water concentration mimicking the initial water content stored at high matric potentials in the smallest pores. The residual bins 1-167 are filled accordingly with particles carrying

the heavy water concentration representing new input water. Further, we also link the three tension areas to bin numbers: bins 1-143 as low-tension area, bins 144-177 as mid-tension area and bins 178-200 as high-tension area. Isotopic concentrations in these tension areas are calculated by averaging concentrations of all particles present in the corresponding bin numbers.

**Table 1.** Experimental and simulation parameters as well as soil hydraulic parameters after van Genuchten (1980) and Mualem (1976) of sandy loam, where $\theta_s$ is the saturated soil water content, $\theta_r$ the residual soil water content, $\alpha$ the inverse of an air entry value, $n$ a quantity characterizing pore size distribution.

| Parameter | Value |
|---|---|
| *Soil type* | Sandy loam |
| $\theta_s$ *(m³ m⁻³)* | 0.41 |
| $\theta_r$ *(m³ m⁻³)* | 0.065 |
| *α (m⁻¹)* | 7.5 |
| *n (-)* | 1.89 |
| *Clay (%)* | 9 |
| *Silt (%)* | 32 |
| *Sand (%)* | 59 |
| $\delta^2$H *(‰) of light water at t = 0 h* | -89 ± 10 |
| $\delta^{18}$O *(‰) of light water at t = 0 h* | -10.8 ± 1.5 |
| $\delta^2$H *(‰) of heavy water at t = 0 h* | -47 ± 1 |
| $\delta^{18}$O *(‰) of heavy water at t = 0 h* | -7.5 ± 0.3 |
| *Low-tension area range (MPa)* | $\sim < 0.016$ |
| | (bins 1-143, $L_D$: 21,000-5,985 µm) |





| | |
|---|---|
| *Mid-tension area range (MPa)* | ~ 0.016 – 1.14<br>(bins 144-177, $L_D$: 5,880-2,415 µm) |
| *High-tension area range (MPa)* | ~ > 1.14<br>(bins 178-200, $L_D$: 2,310-0 µm) |
| *Simulation time step (s)* | 600 |
| *Total number of particles* | $10^5$ |
| *Total number of bins* | 200 |
| *$L_D$ (µm)* | 21,000 |
| *$\delta L_D$ (µm)* | 105 |

### 3.2 Setup of virtual experiment: simulating diffusive pore mixing and leaching of solute during steady state, saturated flow in a soil column

For the virtual experiment, we assume a vertical 1-D soil column of length $z = 1.0$ m, which is subdivided into

vertical layers of $dz = 0.1$ m length (cf. top of Fig 1). The (fully water-saturated) soil column contains the same, macroscopically homogeneous sandy loam with a saturated hydraulic conductivity $K_s$ of $10^{-6}$ m s$^{-1}$, and with all other hydraulic properties and definition of three tension areas in each soil layer as used in the experiment of Bowers et al. (2020). All other experimental and simulation parameters are also the same (cf. Tab. 1). Water particles located initially in the pore space of the surface soil layer (0-0.1 m) carry a concentration $C = 100$ M L$^{-3}$

of a representative conservative solute, while water particles in the other (lower) soil layers carry a zero solute concentration. The soil column is then irrigated by pure water without any solute. A steady-state flow through the soil domain is established for 7 days driven by a free drainage condition at the bottom boundary, neglecting any evaporation effects at the soil-atmosphere interface. For the virtual experiment, the vertical displacement routine of LAST is used, assuming pure matrix flow without the influence of macropores or reactive transport processes

(cf. Sect. 2.1, Zehe and Jackisch, 2016; Sternagel et al., 2019; Sternagel et al., 2021). It calculates a vertical displacement step, by means of advection and dispersion, for each fluid particle in all soil layers, starting from the bottom to the surface layer. Thus, a certain number of particles initially leaves the soil domain via the bottom boundary and to maintain the saturation state, missing numbers of particles in soil layers are gradually refilled by particles from overlying layers until the soil domain is, at the top, finally re-saturated by adding new event particles

to the surface soil layer (steady-state). The length of a vertical displacement step is therefore also dependent on the bin/pore size (cf. Sect. 2.1). Particles in coarse pores experience a larger vertical displacement than particles in smaller pores due to higher advective velocities and diminished capillary effects. Hence, particles in coarse pores/bins are more likely to travel into the next underlying soil layer.

We simulate this virtual experiment setup by our LAST-Model with (i) the DIPMI approach, with constant and

pore-size-distributed diffusivities $D$ (cf. Sect. 2.2) over bins, respectively, and (ii) the common perfect-mixing assumption used in the former versions of LAST (cf. Sect. 2.1, Sternagel et al., 2019; Sternagel et al., 2021)

### 4 Results and Discussion

In Sects. 4.1 and 4.2, we present and discuss the results of the DIPMI simulations of the experiment of Bowers et al. (2020), followed by the presentation and discussion of the results of the virtual experiment in Sects. 4.3 and

30  4.4.





### 4.1 DIPMI simulations of the experiment of Bowers et al. (2020)

Table 2 contains the mean and standard deviation of isotopic concentrations in each tension area and time point for simulation with constant and pore-size-distributed $D$ values, respectively. Values highlighted in bold are within the measured standard deviation range of the experimental values of Bowers et al. (2020), which are additionally

given. It is obvious that most of the simulated isotopic values are in accord with the observation. Deviations mainly occur in the high-tension area after 8 h, as simulations over-predict the degree of mixing (i.e., mixing is too fast) in this high-tension area. All three tension areas require different times to reach a mean equilibrium concentration $\delta_e$ of around -54 ‰ for $\delta^2$H and -8.0 ‰ for $\delta^{18}$O. These equilibrium concentrations are reached in the simulation with pore-size-distributed $D$ values after (i) 8 h to 1 d in the low-tension area, (ii) ~ 1 d in the mid-tension area,

and (iii) 3 d in the high-tension area. Thus, our DIPMI approach simulates complete isotope mixing somewhat faster than Bowers et al. (2020), who found that complete mixing is achieved after around 4 d. However, mixing times and isotopic concentrations of our simulations, with pore-size-distributed $D$ values in particular, are generally consistent with the measured values. Comparing results of simulations with constant and pore-size-distributed $D$ values reveal differences in the mid- and, especially, high-tension areas, while isotopic

concentrations in the low-tension area are quite similar. All tension areas reach the equilibrium state already after 8 h to 1 d when simulating with constant $D$ values in all bins.

**Table 2.** Mean and standard deviation of isotopic concentrations in each tension area and time point for simulation with constant (const.) and pore-size-distributed (distr.) $D$ values, respectively. Bold values are within the standard
deviation range of the measured values of Bowers et al. (2020).

| Tension area | Time point | Mean $\delta^2$H (‰) | | | Mean $\delta^{18}$O (‰) | | |
|---|---|---|---|---|---|---|---|
| | | const. $D$ | distr. $D$ | Bowers | const. $D$ | distr. $D$ | Bowers |
| Low tension | 0 h | **-47 ± 1** | **-47 ± 1** | -47 ± 1 | **-7.5 ± 0.3** | **-7.5 ± 0.3** | -7.5 ± 0.3 |
| ~ < 0.016 MPa | 8 h | **-53 ± 2** | **-50 ± 2** | -53 ± 1 | **-8.0 ± 0.5** | **-7.7 ± 0.4** | -7.8 ± 0.2 |
| | 1 d | **-54 ± 3** | **-53 ± 3** | -56 ± 1 | **-8.0 ± 0.5** | **-8.0 ± 0.5** | -8.0 ± 0.2 |
| | 3 d | **-54 ± 3** | **-54 ± 3** | -56 ± 1 | **-8.0 ± 0.5** | **-8.0 ± 0.5** | -7.8 ± 0 |
| | 7 d | **-54 ± 3** | **-54 ± 3** | -55 ± 1 | **-8.0 ± 0.5** | **-8.0 ± 0.5** | -7.3 ± 0.3 |
| Mid tension | 0 h | **-59 ± 4** | **-59 ± 4** | -65 ± 4 | **-8.5 ± 0.7** | **-8.5 ± 0.7** | -9.2 ± 0.6 |
| ~ 0.016 – 1.14 | 8 h | **-56 ± 4** | **-61 ± 4** | -63 ± 5 | **-8.3 ± 0.5** | **-8.6 ± 0.7** | -8.6 ± 0.4 |
| MPa | 1 d | -55 ± 3 | **-57 ± 3** | -60 ± 0 | **-8.1 ± 0.5** | **-8.3 ± 0.5** | -8.3 ± 0.2 |
| | 3 d | **-54 ± 2** | **-55 ± 3** | -57 ± 1 | **-8.0 ± 0.5** | **-8.1 ± 0.6** | -7.9 ± 0.2 |
| | 7 d | **-54 ± 2** | **-54 ± 2** | -55 ± 0 | -8.0 ± 0.5 | -8.1 ± 0.6 | -7.0 ± 0.2 |
| High tension | 0 h | **-89 ± 10** | **-89 ± 10** | -89 ±10 | **-10.8 ± 1.5** | **-10.8 ± 1.5** | -10.8 ± 1.5 |
| ~ > 1.14 MPa | 8 h | -57 ± 3 | **-71 ± 6** | -79 ± 3 | -8.3 ± 0.5 | **-9.4 ± 1.0** | -9.5 ± 0.4 |
| | 1 d | -55 ± 3 | -60 ± 4 | -72 ± 4 | -8.1 ± 0.5 | **-8.5 ± 0.7** | -8.4 ± 0.2 |
| | 3 d | -54 ± 3 | -55 ± 3 | -65 ± 2 | -8.0 ± 0.5 | **-8.1 ± 0.5** | -7.6 ± 0.6 |
| | 7 d | -54 ± 3 | -54 ± 3 | -62 ± 2 | -8.0 ± 0.5 | -8.1 ± 0.5 | -6.5 ± 0.5 |

Figure 2 shows additionally the isotopic concentrations, simulated with pore-size-distributed $D$ values and the observations, for each tension area and time point in a dual-isotope space. The isotopic concentrations in the three tension areas gradually converge towards the mean equilibrium concentrations over time. Simulated
concentrations in low- and mid-tension areas are in accord with the observations, indicated by the overlapping of simulated and measured standard deviation ranges. An exception is the high-tension area, where measured isotopic concentrations of especially $\delta^2$H have lower values compared to our simulations after 8 h; these findings show that





the actual mixing process in the smallest pores is delayed. Further, there are also differences between the two isotope species. For $\delta^{18}$O, simulated concentrations are consistent with the observations in all tension areas over the entire period, except the 7 d concentrations in the mid- and high-tension areas. However, these measured concentration values were unexpectedly high in the experiments of Bowers et al. (2020) as further discussed in

Sect. 4.2. The value range of $\delta^{18}$O is also generally larger compared to $\delta^2$H.

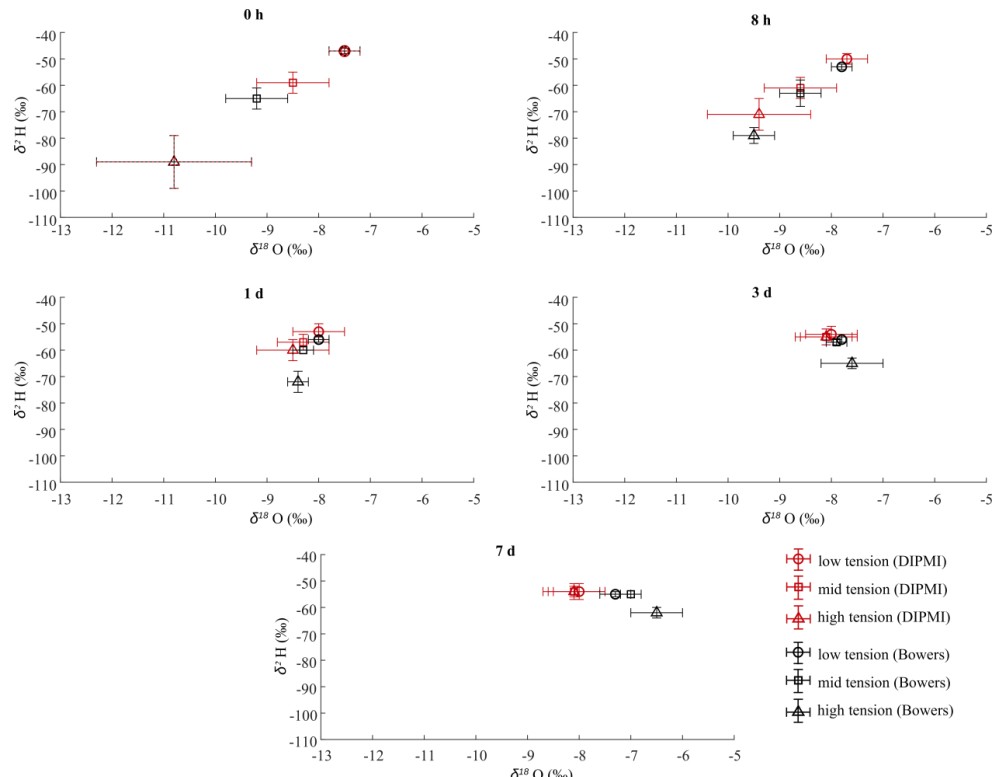

**Figure 2.** Isotopic concentrations in dual-isotope space for each tension area and time point, simulated (red) by the DIPMI approach with pore-size-distributed $D$ values and measured (black) by Bowers et al. (2020). Note that
at t = 0 h the simulated and measured isotopic values in the low- and high-tension areas are identical.

**4.2 Analysis of simulations of the experiment of Bowers et al. (2020)**

The results of simulating the experiment of Bowers et al. (2020) (cf. Sect. 4.1) indicate that our diffusive pore mixing (DIPMI) approach is capable to reproduce the experiment of Bowers et al. (2020). The results in Sect. 4.1 show that our approach is suitable to (i) simulate the measured mixing time (~ 3 d) required to reach an equilibrium
concentration $\delta_e$ in the entire pore space and (ii) resolve most of the isotopic concentrations in all tension areas with time (cf. Sect. 4.1, Tab. 2 and Fig. 2).

In general, the concept of DIPMI is consistent with studies, which usually observe a lag time in isotopic mixing of waters stored initially in different pore sizes (e.g., Adams et al., 2020; Bowers et al., 2020; Orlowski and Breuer, 2020), proving that diffusive mixing over an entire pore space is far from being a perfect, instantaneous process,
even on the (small) scale of a few centimetre long soil sample. A realistic description of diffusive mixing processes





is especially crucial for interpreting studies examining origins of plant water (e.g. Sprenger et al., 2016; Penna et al., 2020), water ages (e.g. Hrachowitz et al., 2013; Sprenger et al., 2019), travel times (e.g. Klaus et al., 2015; van der Velde et al., 2015) or even groundwater (e.g. Berkowitz et al., 2016).

Our calculated extent of $L_D = 21,000$ µm (cf. Eqs. 1 and 2) appears to feasibly resolve the internal pore space of

the sandy loam soil volume used in the experiment of Bowers et al. (2020). The distributed, bin-dependent diffusivities $D$ (cf. Eq. 4) facilitate a realistic simulation of measured isotopic concentrations, which is superior to simulations with constant $D$ values (cf. Tab. 2). The latter results in over-prediction of the degree of mixing with reaching the equilibrium state almost simultaneously in all tension areas after just 8 h to 1 d. A constant diffusivity of $2.272 \cdot 10^{-9}$ m² s⁻¹ seems feasible only in coarser pores. The smaller a pore size class the greater the capillary

forces and friction, caused by interactions between solid surfaces and the thin water layers directly adjacent to them. Both, capillary forces and friction, significantly decrease water movement in the smallest pores. This effect corroborates our calculation of pore-size-dependent diffusivities. However, $D$ values in the high-tension area are still too high because measured concentrations are not matched in the same extent as it is the case in the low- and mid-tension area. This entails that our approach still simulates overly strong mixing in the high-tension area.

The high-tension area of the water retention curve probably bears the highest uncertainty. In this area of the pore space, water is held at tensions >> 1.0 MPa and is usually extracted by cryogenic vacuum distillation (CVD). This method, however, may be prone to produce artefacts when analysing isotopic concentrations of soil and plant water (Orlowski et al., 2013; Adams et al., 2020; Orlowski and Breuer, 2020; Allen and Kirchner, 2021). Beside possible uncertainties in the measurement procedure, specific subscale processes can be the reason for discrepancies

between simulation and observation in the high-tension area. Bowers et al. (2020) suggested that interactions/adsorption of water ions with clay minerals within smaller pores can have a significant effect on the mixing behaviour. This might be also the reason for the discrepancies between simulated and measured $\delta^{18}O$ concentrations in mid- and high-tension areas after 3-7 d, as measured values are higher than expected as stated by Bowers et al. (2020). Such $\delta^{18}O$ enrichment was also reported in other studies (e.g., Oerter et al., 2014).

Orlowski and Breuer (2020) further suggested that isotopic fractionation may occur during diffusive mixing, especially in high-tension areas. Reasons for such isotopic fractionations are difficult to define and in addition to adsorption effects, further subscale processes may play a role, such as adhesion or evaporation effects. Evaporation is often regarded as a main driver for isotopic fractionation, especially at the interface of the soil-atmosphere system (Sprenger et al., 2016; Sprenger et al., 2018). However, it may also result in a fractionation effect on the

pore scale during the water extraction process in experimental studies when a phase change from liquid to gaseous occurs at the interface of saturated and unsaturated pores, which in turn may lead to vapour-pressure-controlled adsorption of water on soil surfaces (Lin and Horita, 2016; Lin et al., 2018).

Such detailed subscale processes are not incorporated specifically in the current version of our DIPMI approach. We focus on the abstraction of physical properties, which we think have generally the greatest influence on the

diffusion process inside of soil domains, e.g., diffusion coefficient of certain fluid or substance, distinct water pockets in soil pores of different sizes, tortuosity of flow paths (Chou et al., 2012; Bowers et al., 2020). We lump these physical properties into the two main assets of our DIPMI approach: (i) the extent and characteristic flow length of the pore space $L_D$, and (ii) variable diffusivities $D$ in different pore size classes, both of which can be derived directly from the soil water retention curve (cf. Sect. 2.2).





Results of Bowers et al. (2020) and our simulations both highlight that mixing processes in soils are by no means instantaneous or perfect, even in very small and fully saturated control volumes. Rather, diffusive spreading of water depends on the pore size distribution and specific soil water retention properties. With this insight, it is of interest to examine, in the following virtual experiment (Sects. 4.3 and 4.4), how pore-size-dependent and non-

instantaneous mixing affect simulations of water flow through a saturated soil column on a larger scale, and to delineate their effects on solute breakthrough and redistribution in soil.

### 4.3 Simulation of the virtual experiment

Solute breakthrough curves exhibit a clear difference between simulations with the DIPMI approach and the perfect-mixing assumption (Fig. 3). Simulation with the perfect-mixing assumption results in a breakthrough curve

that is shaped like an approximately normal distribution with a concentration peak of 13 M $L^{-3}$ after ~ 42 h, followed by a sharp decrease converging a zero solute concentration after ~ 105 h. Thus, all solute stored initially in the surface soil layer is eluted completely from the entire soil column by ~ 2.3 pore volumes, when simulation is performed with the perfect-mixing assumption. In contrast, simulation with the DIPMI approach and constant $D$ values results in a breakthrough curve that exhibits a right-skewed distribution. The peak concentration of ~ 7

M $L^{-3}$ is reached after around 38 h and is followed by a long tailing of solute breakthrough, which converges the zero concentration only after 7 days, corresponding to 3.5 pore volumes. Hence, significantly more time is required to elute all solute from the soil column. The simulation with the DIPMI approach and pore-size-distributed $D$ values generally results in a breakthrough curve with a similar shape, as the breakthrough curve with constant $D$ values, with comparable peak concentration and long tailing behaviour. However, it shows a shorter time to peak

(~ 22 h) and the solute breakthrough tailing does not converge to zero concentration at all, not even after the simulation period of 7 days. Thus, more than 3.5 pore volumes are needed to leach all solute from soil in the case of pore-size-distributed $D$ values.

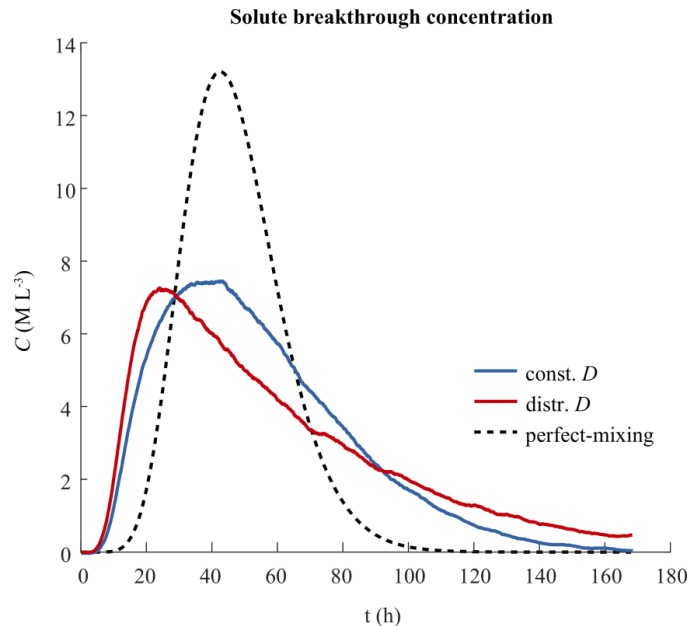

**Figure 3.** Solute breakthrough curves over 7 d, simulated with (i) the DIPMI approach and constant $D$ (const.) values (blue), (ii) DIPMI approach and pore-size-distributed $D$ (distr.) values (red), as well as (iii) the perfect-mixing assumption (black).

Figure 4 further shows the mean solute concentrations in the three tension areas per vertical soil layer at different points in time. Comparing the results of simulations with the DIPMI approach (with pore-size-distributed $D$ values) and the perfect-mixing assumption generally supports the previous insights of the breakthrough curves.

There is no difference in concentration between tension areas when using the perfect-mixing assumption, as this

10 approach averages solute concentrations over the entire pore space of a soil layer (cf. Sect. 2.1). Consequently, solute gradually propagates, in the form of a fast wave, through all soil layers resulting in a complete elution of solute in all soil layers within the first 3-4 days. Results of the simulation with the DIPMI approach and pore-size-distributed $D$ values exhibit a different and more heterogeneous picture (red shaded symbols and lines), especially regarding solute propagation behaviour and concentrations in different tension areas. Due to the imperfect mixing,

15 a large fraction of solute preferentially propagates vertically through the low-tension areas (dark red) of the soil layers. This vertical leaching process is faster than the diffusive mixing over the tension areas within a soil layer, leading to the fast breakthrough of solute during the first 12 h (cf. Fig. 3). Hence, only smaller amounts of solute enter the mid- and high-tension areas (light red and orange) in deeper soil layers (0.1-1.0 m). In contrast, the initially solute-filled mid- and high-tension areas in the surface soil layer only slowly release their solute. This

20 mechanism entails a retardation effect, which results in the persistence of solute in soil over the entire period of 7 d and the long breakthrough curve tailing with incomplete elution of solute (cf. Fig. 3).



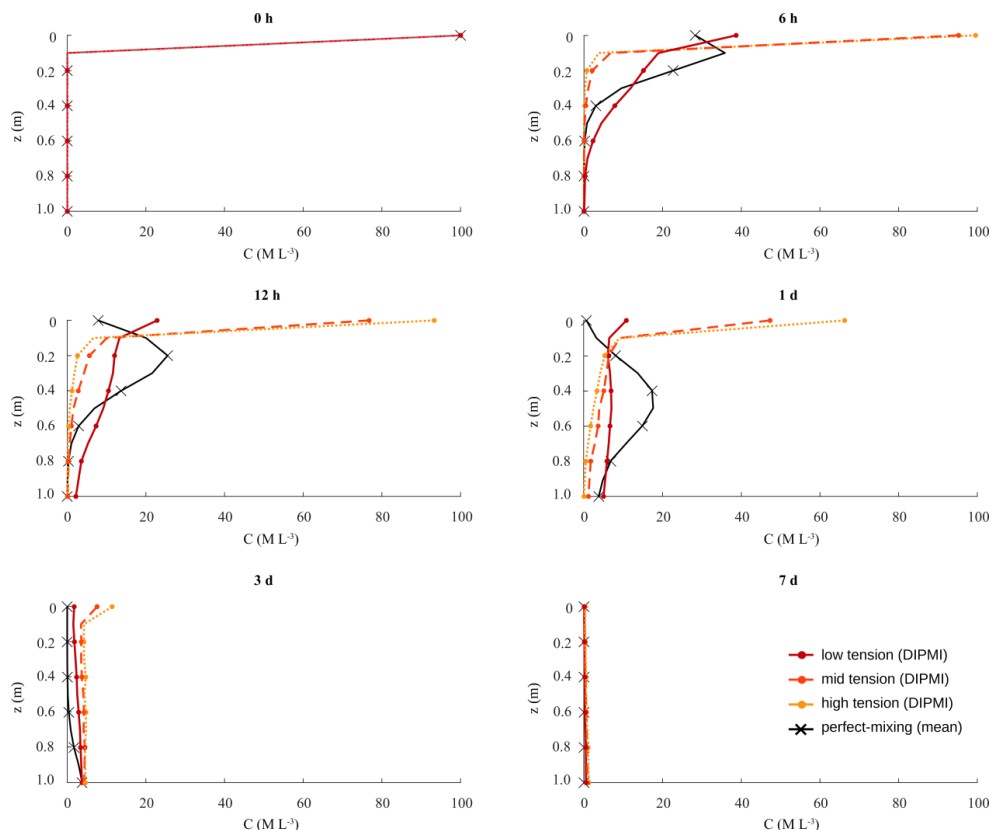

**Figure 4.** Vertical concentration profiles of a solute with mean concentrations in three tension areas over 7 d, simulated by the DIPMI approach with pore-size-distributed $D$ values (red shades) and the perfect-mixing assumption (black), respectively. Note that the black line and crosses illustrate the same mean concentration in all three tension areas per soil layer, as the perfect-mixing assumption averages out pore-size-dependent differences.

### 4.4 Analysis of the virtual experiment

Use of the perfect-mixing assumption shows two effects in our virtual experiment: (i) a longer time to first breakthrough and peak (cf. Fig. 3), as well as (ii) a steeper and shorter tailing of solute breakthrough concentration after the peak with a complete leaching of solute within 105 h (cf. Fig. 4). These effects can be explained by the fundamental assumption that solutes always perfectly and instantaneously mix over the entire pore space in each soil layer during passage of the soil domain. Consequently, solute uniformly spreads over the pore spaces of initially solute-free soil layers (0.1-1.0 m) before reaching the bottom boundary, which leads to dilution and the delayed breakthrough front arrival. Thereafter, the perfect and instantaneous mixing of pure infiltrating event water with solute-containing, pre-event water causes the rapid and complete elution of all solute, especially out of the surface soil layer, also with a relatively high peak concentration. These dilution and elution effects are characteristic of over-mixing phenomena (e.g., Green et al., 2002; Boso et al., 2013) and may be problematic for, e.g., assessing the risk of contaminant leaching by potentially giving wrong predictions regarding breakthrough times and persistence in soil.





Using the DIPMI approach with pore-size-distributed $D$ values (cf. Eq. 4) for simulation of the virtual experiment bears opposite effects: (i) a faster initial breakthrough (cf. Fig. 3), and (ii) a long tailing of solute breakthrough concentration with an incomplete elution of solute (cf. Fig. 4). The timescale for diffusive mixing over pore sizes is significantly larger than the timescale for vertical transport of solute. This leads to an early arrival of the

breakthrough front, as solute travels downward mainly through the pores of low-tension areas without spreading uniformly over soil layers. The smaller peak concentration and long tailing of the breakthrough curve are, thereafter, caused by a retardation effect of pores in the mid- and high-tension areas. These pores, with smaller diffusivities and higher capillary tensions, bind solute for a longer time (cf. Fig. 4) before they are mixed diffusively with pure infiltrating water and leached out. Regarding these effects, it is important to recall that the

vertical displacement of fluid particles depends also on the different pore size classes/bins in our LAST-Model (cf. Sect. 2.1, Sternagel et al., 2019; Sternagel et al., 2021). Particles in coarse pores of low-tension areas are more mobile and displaced vertically by a higher advective velocity compared to particles in smaller pores. Consequently, water and solute flow mainly through these pores in low-tension areas, with a limited diffusive exchange with smaller pores and thus, the mid- and high-tension areas are essentially bypassed. This behaviour

implies that the saturated flow system is dominated by preferential bypassing flow through the low-tension area. In the low-tension area, the mean $D$ values of the constant $D$ method ($2.272 \cdot 10^{-9}$ m² s$^{-1}$) and the pore-size-distributed $D$ method ($1.243 \cdot 10^{-9}$ m² s$^{-1}$) are (relatively) similar, leading to breakthrough behaviours of both methods that are comparable in an early phase of breakthrough (cf. Fig. 3). However, the overall higher and constant $D$ value causes a stronger leaching of retarded solute over time in later phases of breakthrough, especially

from mid- and high-tension areas, with a complete elution after the period of 7 d. This implies that during early phases of fast, bulk leaching of solute the influence of pore-size-dependent diffusive mixing is less significant due to preferential bypassing. Nevertheless, pore-size-dependent diffusive mixing becomes highly relevant in later phases of breakthrough, when residual amounts of solute, stored and retarded in small pores, are gradually moved back into coarser pores by diffusive mixing with infiltrating water and hence, remobilized. Note that we perform

our simulations in saturated media because in a fully saturated pore space, differences of the diffusivities between the largest, saturated pore size class and the smallest, saturated pore size class are more distinct than in a partially saturated pore space. Thus, simulations under saturated conditions are more suitable to highlight the influence of diffusive mixing with pore-size-distributed $D$ values, also in comparison to the use of constant $D$ values and the perfect-mixing assumption.

Overall, results of the virtual experiment reveal a different behaviour of solute leaching and redistribution in soil for simulation with the DIPMI approach, compared to simulations invoking the perfect-mixing assumption. In particular, the long tailing of breakthrough curves is in line with common observations of several studies (e.g., Zinn and Harvey, 2003; Willmann et al., 2008; Edery et al., 2014). Zinn and Harvey (2003) linked the long tailing of breakthrough curves to mass transfer between regions of different mobility, e.g., pore size classes in different

tension areas. Edery et al. (2014) also simulated solute breakthrough in saturated, porous media by a Lagrangian particle-tracking approach. They showed that the broadness and tailing of breakthrough curves increase with a generally higher heterogeneity of pore space and flow paths. Our subdivision of the pore space into different pore size classes with pore-size-dependent diffusivities in the DIPMI approach is in line with this finding and adds, furthermore, an important aspect to account for imperfect subscale mixing in soil-hydrological modelling.


Moreover, our results highlight that these typical shapes of breakthrough curves are not exclusively caused by explicit hydraulic (e.g., macropore flow) or chemical (e.g., adsorption, desorption) heterogeneity in soil, but that early breakthroughs and long tailing may also be a result of imperfect diffusive mixing within the pore space (Willmann et al., 2008), even when the flow domain is fully saturated and its soil properties are macroscopically

5   homogeneous. Hence, we can verify our proposed hypothesis (cf. Sect. 1) and state that imperfect mixing across soil pores with different hydraulic properties within the soil matrix may entail an implicit heterogeneity of flow causing typical shapes of breakthrough curves, even in the absence of explicitly defined preferential flow paths.

## 5 Conclusions and Outlook

The main findings of this study are as follows:

- Simulation results show the feasibility of the DIPMI approach to reproduce the findings of the experiment of Bowers et al. (2020), by capturing correctly the measured diffusive mixing times and concentrations of water isotopes over the pore space of a fully saturated soil sample (cf. Sect. 4.1, 4.2).
- A virtual experiment highlights that imperfect mixing in a macroscopically homogeneous soil matrix can

15         produce breakthrough patterns that are usually associated with preferential flow in heterogeneous soil structures such as macropores (cf. Sect. 4.3, 4.4).
- The DIPMI approach is physically-based in the sense that its model parameters can be derived from the soil water retention curve and no further calibration is needed (cf. Sect. 2.2).

20   In the future, in-situ 1-D column experiments are planned to analyse the influence of the microstructure of partially saturated soils on the temporal and spatial mixing of isotopes over the pore space. The experimental results may provide a representative dataset serving as a reference for further extending and testing of the DIPMI approach, as comparable data on pore-scale mixing processes remain scarce.



*Data availability.* A previously published version of the LAST-Model (Sternagel et al., 2019) is available in a GitHub repository: https://github.com/KIT-HYD/last-model (Mälicke and Sternagel, 2021). We intend to provide the code of the DIPMI approach and relevant data in this repository, too. Otherwise, please contact Alexander Sternagel (alexander.sternagel@kit.edu).

*Author contributions.* AS wrote the paper, did the main code developments and carried out the analysis. RL, BB and EZ contributed to the theoretical framework as well as helped with interpreting and editing.

*Competing interests.* The authors declare that they have no conflict of financial competing interest. However, 10 authors EZ and BB are members of the editorial board of the journal.

*Acknowledgements.* We thank William H. Bowers and his colleagues for performing their interesting experiment and providing all data on a freely accessible database. Our work was inspired significantly by their study.

15 *Financial support.* The article processing charges for this open-access publication were covered by a Research Centre of the Helmholtz Association.





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
