# Peer review of "Stepping beyond perfectly mixed conditions in soil hydrological modelling using a Lagrangian approach"

_Hydrology and Earth System Sciences, 2021_

## Author Comment (AC1)

**Response to comments of Anonymous Referee #1**

On behalf of all co-authors, I sincerely thank the Anonymous Referee #1 for the thoughtful and detailed assessment of our work.

*R1: This manuscript is a novel contribution to studying diffusive processes in soil. A Lagrangian model is developed to account for heterogeneity in diffusive mixing in soil caused by pore structure. The pore size distribution is accounted for through varying diffusion coefficients. The model has first been tested by comparing it to the recent experiments on effectively non-Fickian diffusive mixing and shown to agree well in terms of time and concentration. Further, simulations were designed to combine diffusion and leaching in a saturated soil column and demonstrate the non-Fickian transport behaviour in this system. While the impact of physical or pore structure heterogeneity has commonly been shown to result in non-Fickian transport, this manuscript systematically increases the level of complexity in terms of diffusive processes studied and shows an interesting workflow to characterize and quantify the non-linear transport behaviour. I can therefore recommend it for publication after the following minor suggestions were considered and addressed.*

**AS:** Thank you very much for your positive assessment of our work. We really appreciate that you see our approach as a novelty and a step forward towards a more complex and realistic description of subscale diffusion processes in hydrological models.

**Main comments**

*R1: Pore space and water retention curve are subdivided using N=200. Did you investigate sensitivity to this number of bins?*

**AS:** In this study, we do not specifically investigate the sensitivity of $N$. However, Zehe and Jackisch (2016) already used a similar Lagrangian approach to simulate soil water dynamics, on which the presented DIPMI approach is based. They did an analysis of the sensitivity of $N$ and found that $N > 50$ is favourable to produce good simulation results compared to a Richards solver. Further, Talbot and Ogden (2008) used a comparable method to discretize the soil moisture domain into bins and they found that $N = 200$ is a sufficient number that can be applied for most soil types. We will add this information to our revised manuscript.

*R1: In relation to the previous question how is the bin size related to pore or throat radius that could have been actually measured in this type of soil?*

**AS:** The soil water retention curve corresponds, due to the Young-Laplace equation (Eq. 1 in manuscript), to the cumulative probability density distribution of pore sizes. Particles in a bin represent the contribution of the water storage in a small range of pores size classes to the overall water storage in the entire pore space. Each pore size class is defined by a specific relation between matric potential and water content, which can be derived from the soil water retention curve, e.g., of the sandy loam measured by Bowers et al. (2020). With this measured matric potential – water content relationship, it is then possible to make a connection to corresponding pore radii of each pore size class by using the Young-Laplace equation. From the given soil water retention curve in the study of Bowers et al. (2020), you can infer pore diameters corresponding to the respective matric potential ranges of the sandy loam. We will include a comment on this in the revised manuscript.

*R1: Can you comment on the expected impact of dimensionality and anisotropy?*

**AS:** Thank you for this interesting point. In the recent version, our model approach is limited to one dimension and anisotropy is not considered. Until now, we have been applying our model on rather small spatial scales up to a few m², but we think that dimensionality and anisotropy especially have an impact on larger, more heterogeneous scales. However, increasing the dimensionality, e.g., with a second flow dimension in the horizontal x direction, is generally possible to implement into our approach. Jackisch and Zehe (2018) implemented this successfully with a similar particle-based Lagrangian approach, but due to the higher model complexity with a necessarily higher number of particles, simulation times increased excessively. Thus, we suggest it is necessary to first develop suitable assumptions and simplifications to extend our model approach to higher dimensions in a meaningful manner.

The same is true for anisotropy. Currently, we assume homogeneous, isotropic soil properties over a flow domain in our simulations. Our model, however, gives in general the opportunity to specify freely the properties in each vertical soil layer and even in each bin/pore size class within a soil layer. In this way, it would be able to integrate anisotropic conditions by, e.g., depth-dependent soil properties with different pore size distributions in each layer and thus, also various hydraulic and diffusive characteristics for flow in different directions of the flow domain.

*R1: I would leave this to the authors to decide but to me the term "virtual experiment" is quite confusing. Why not simply say that you run your simulations to explore the model predictive capability in a more complex setting after you have demonstrated a good agreement with the experiment.*

**AS:** Thank you and sorry for the confusion. You summed up correctly the purpose of the "virtual experiments". With this term, we want (i) to give these simulations a specific name to which we can refer throughout the manuscript, and (ii) to make clear that these simulations are not based on real-existing experiments. I think we will maintain the term "virtual experiment" or at least a similar term, but we will additionally explain it in 1-2 sentences to clarify its meaning.

Minor comments

*R1: Line 22 instead of "… comparing…" it should read as "…compared…"*

**AS:** Thanks. We will check this sentence accordingly.

Thank you very much,

Alexander Sternagel on behalf of all authors

**References**

Bowers, W. H., Mercer, J. J., Pleasants, M. S., and Williams, D. G.: A combination of soil water extraction methods quantifies the isotopic mixing of waters held at separate tensions in soil, Hydrology and Earth System Sciences, 24, 4045-4060, 2020.

Jackisch, C., and Zehe, E.: Ecohydrological particle model based on representative domains, Hydrology And Earth System Sciences, 22, 3639-3662, 10.5194/hess-22-3639-2018, 2018.

Talbot, C. A., and Ogden, F. L.: A method for computing infiltration and redistribution in a discretized moisture content domain, Water Resources Research, 44, 2008.

Zehe, E., and Jackisch, C.: A Lagrangian model for soil water dynamics during rainfall-driven conditions, Hydrology And Earth System Sciences, 20, 3511-3526, 10.5194/hess-20-3511-2016, 2016.

---

## Author Comment (AC2)

**Response to comments of Ehsan Ranaee**

On behalf of all co-authors, I sincerely thank Ehsan Ranaee for his thoughtful and detailed assessment of our work.

*ER: Dear Editor, Thank you for sharing this manuscript with me.*

*Authors extended their previously developed Lagrangian Soil Water and Solute Transport (LAST) model; and presented a novel modeling approach of diffusive pore mixing (DIPMI). This modeling strategy is implemented for simulating reactive solute transport in partially water saturated soil domains. A key development of this modeling approach (with respect to their farmer LAST model) is that DIPMI gets rid of assuming perfect mixing of solutes among water particles. To this end, DIPMI is developed to account for the self-diffusion of water particles across a characteristic length scale of the pore space using pore-size-dependent diffusion coefficients.*

*Authors tested DIPMI approach to reproduce some experimental findings (from the literature) with diffusive mixing of water isotopes over the pore space of a fully saturated soil volume. They also performed simulation of mixing of a representative solute in a vertical - saturated soil column and compare results of the DIPMI approach against the ones that employed common perfect-mixing assumption.*

*Authors suggest that imperfect mixing of water and solutes in the pore space can result long tailing of corresponding solute breakthrough curves which agrees with experimental outcomes. Indeed, solute breakthrough curves of the simulations with the LAST approach of perfect-mixing assumption may exhibit clear differences to experimental outcomes.*

*This work can provide some insights to the simulation of imperfect subscale mixing in a macroscopically homogeneous soil matrix.*

*The original idea sounds interesting to me, and I believe this paper can be published in HESS with very mild revisions.*

**AS:** Thank you very much for your positive assessment of our work. In line with our response to Anonymous Referee #1, we appreciate that the scope and intention of our work, as well as the provided insights into simulating diffusive mixing on the pore scale, are all clear to the reader.

*ER: In particular, I would suggest Author to extend the work by evaluating sensitivity of the simulation responses to the variation of the molecular diffusion coefficient values.*

**AS:** Thanks for your interesting suggestion. In fact, we already performed a kind of sensitivity analysis by determining the breakthrough curve simulations (cf. Figure 3) with significantly different diffusion coefficients $D$, in the setups with distributed ($\triangleq$ mean: $9.7 \cdot 10^{-10}$ m²/s) and constant ($\triangleq$ mean: $2.29 \cdot 10^{-9}$ m²/s) $D$ values ($\triangleq$ range of 58 %), as well as with the instantaneous, perfect-mixing assumption ($\triangleq$ infinite diffusion coefficient: $10^{\infty}$ m²/s). Further simulations with (i) higher/faster diffusion coefficient values will result in breakthrough curves with shorter tailings gradually approaching the shape of the curve of the perfect mixing assumption, and (ii) even smaller/slower diffusion coefficient values will result in breakthrough curves with increasingly longer tailings. Hence, we think that further sensitivity simulations would not yield significantly more insights. However, we will add a statement about the sensitivity of diffusion coefficients to the discussion, to explicitly address the point raised here.

*ER: Legends and axis of Figure 4 are hardly visible.*

**AS:** Thank you for this point. We will increase the font size of the axis and legends to enhance the visibility of Figure 4.

Thank you very much,

Alexander Sternagel on behalf of all authors